# Learning Overlap Detection for Domain-Adaptive Image-to-Point Cloud Registration

## Abstract

Outdoor registration methods often employ an dedicated module to detect overlapping regions between images and point clouds. While effective, this strategy is not directly applicable to indoor scenarios and increases computational cost. However, to further improve indoor registration accuracy, it is crucial to identify and isolate overlapping regions, minimizing interference from non-overlapping areas. Furthermore, without targeted design, aligning image and point cloud features may lead to mismatches during feature interaction. To address these issues, we propose two modules: the Reinforcement Learning Overlap Detector (RLOD) and the Hierarchical Domain Adaptation Interaction (HDAI) module. RLOD adaptively selects overlapping regions by leveraging intrinsic geometric information, thus constraining the matching space and improving accuracy. HDAI aligns image and point cloud features at both mean and covariance levels, mitigating cross-modal discrepancies and stabilizing attention. Experiments on RGB-D Scenes v2 and 7-Scenes benchmarks demonstrate that our method achieves superior performance, setting a new state of the art for image-to-point cloud registration.

## 1 Introduction

Image-to-point cloud registration (I2P) aims to determine the rigid transformation between the coordinate systems of a point cloud and a camera capturing the same scene. This process underpins many vision tasks such as 3D reconstruction Mouragnon et al. (2006); Deng et al. (2024), SLAM Durrant-Whyte & Bailey (2006); He et al. (2023), and localization Bolognini et al. (2005); Wu et al. (2024). Yet images are dense, structured 2D grids, while point clouds are sparse, unordered 3D sets, leading to a substantial domain gap in their feature distributions. Additionally, due to the distinct sensing characteristics of different sensors, the spatial coverage of images and point clouds is often not perfectly aligned, leading to non-overlapping or missing regions. As a result, effectively modeling and aligning their representations in the overlapping areas remains a challenging problem.

For the problem of I2P, different strategies have been developed for indoor Feng et al. (2019); Wang et al. (2021); Ren et al. (2022) and outdoor Kang et al. (2023); Yue et al. (2025) scenarios due to the distinct sensing modalities and viewpoints involved. In indoor scenarios, 2D3D-MATR Li et al. (2023) introduced the first coarse-to-fine, detection-free framework. It establishes patch-level correspondences between image and point cloud features, progressively refines them into pixel-to-point matches, and finally estimates the rigid transformation using PnP-RANSAC Lepetit et al. (2009); Fischler & Bolles (1981). In contrast, outdoor methods, such as ICL-I2P Li et al. (2025), utilize an overlap region detection module. This module leverages the similarity between high-dimensional image and point cloud features to identify overlapping areas and obtain an initial pose. Within these detected overlap regions, keypoint correspondences are extracted for pose refinement, showing significant potential. However, in indoor scenes, where the overlap between images and point clouds is generally substantial (see Figure 1(a)), relying on global image features for overlap detection often leads to suboptimal results. The additional detection module also substantially increases computational complexity. This leads us to explore whether the concept of constructing overlap regions, as seen in outdoor methods, can be redesigned in a more lightweight manner and adapted for indoor registration. This adaptation could enable more efficient and accurate performance improvements.

Through the above discussion of registration methods, we summarize two key issues that need to be improved in order to accurately and reliably accomplish indoor registration. *1. How to design*

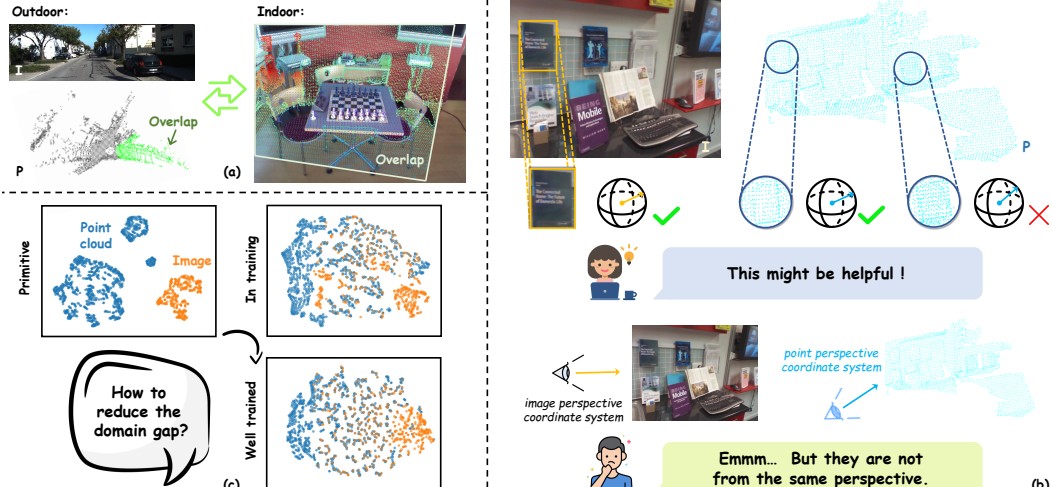

Figure 1: (a) Comparison of overlap region ranges in outdoor and indoor registration tasks. (b) Illustration of the reasoning process for exploring intrinsic geometric information from normal. (c) t-SNE visualization of image and point cloud features under different conditions.

***a lightweight overlap detection module for indoor I2P to enhance accuracy and robustness.*** For overlap region detection, we aim to incorporate more 3D information to optimize the judgment. Depth maps are a common method for obtaining 3D information from images, but their inherent scale ambiguity makes them challenging to use directly for local feature matching. In contrast, surface normals are invariant to translation and scale, which makes them more suitable as geometric features for registration tasks. However, since images and point clouds are captured from different viewpoints (Figure 1(b)), it may be beneficial to extract intrinsically invariant geometric structures to improve overlap region determination. Moreover, as overlap detection varies across scenes, enabling the model to adaptively select potential regions can further enhance accuracy and robustness.

***2. How to align image and point cloud features to ensure consistent representation within the overlapping regions.*** As shown in Figure 1(c), the substantial differences between 2D and 3D data lead to inconsistent feature spaces. Since transformer attention relies on query–key similarity, large distribution gaps between image and point cloud keys cause the same query to yield divergent attention across modalities, resulting in unstable cross-attention alignment. Through a tailored design, cross-modal multi-level attention might be maintained on a comparable numerical scale to enhance stability, while preserving modality-specific information, thereby enabling more accurate and robust registration of overlapping regions.

To address the above challenges, we propose a novel method, Learning Overlap Detection for Domain-Adaptive Image-to-Point Cloud Registration, which introduces two innovative modules: the Reinforcement Learning Overlap Detector (RLOD) and the Hierarchical Domain Adaptation Interaction Module (HDAI). In RLOD, we leverage surface normals to enrich images with 3D cues. Surface normal labels are derived from depth maps Yang et al. (2024) for images and directly computed from 3D data for point clouds. However, as they originate from different coordinate systems, viewpoint variations cause inherent discrepancies. To address this, we exploit intrinsic geometric information to strengthen correlations between image and point cloud. In addition, a reinforcement learning–driven strategy enables the model to adaptively select overlapping regions, achieving more accurate and efficient detection while reducing false matches. In HDAI, we enhance the transformer by aligning image and point cloud features at both mean and covariance levels. This statistical alignment mitigates cross-modal discrepancies, stabilizes attention computation, and alleviates drift from scale mismatches, thereby improving the robustness and generalization of cross-modal fusion.

In summary, our contributions are:

- We propose a novel method, Learning Overlap Detection for Domain-Adaptive Image-to-Point Cloud Registration, achieving excellent accuracy and strong generalization in cross-modal registration.
- We design a Reinforcement Learning Overlap Detector that leverages intrinsic geometric properties to enhance correlation and adaptively detect overlapping regions. We further introduce a

Hierarchical Domain Adaptation Interaction module to align feature distributions via hierarchical cross-modal interactions, alleviating attention drift and improving representation consistency.

• Extensive experiments and ablations on RGB-D Scenes v2 and 7-Scenes demonstrate the superiority of our network, setting a new state of the art in image-to-point cloud registration.

## 2 RELATED WORKS

In this section, we briefly review related works on image-to-point cloud registration, covering stereo image registration, point cloud registration, and inter-modality registration.

**Stereo Image Registration.** Traditional stereo image registration was dominated by detector-based methods. Before deep learning, handcrafted techniques such as SIFT Ng & Henikoff (2003) and ORB Rublee et al. (2011) were widely used to detect keypoints and establish 2D matches from local descriptors. With the advent of deep learning, neural network–based detectors significantly advanced this field. A notable milestone is SuperGlue Sarlin et al. (2020), which introduced Transformer-based Vaswani et al. (2017) feature matching and substantially improved local correspondence quality. However, the difficulty of detecting repeatable keypoints in textureless or non-salient regions has motivated detector-free approaches. Recent methods such as LoFTR Sun et al. (2021) and Efficient LoFTR Wang et al. (2024) employ coarse-to-fine pipelines with Transformers, enabling dense matching with global receptive fields.

**Point Cloud Registration.** Point cloud registration methods have evolved from handcrafted descriptors, such as PPF Moheimani et al. (2006) and FPFH Rusu et al. (2009), to learning-based techniques. CoFiNet Yu et al. (2021) was among the first to introduce a detector-free coarse-to-fine framework for registration. More recently, traditional RANSAC Fischler & Bolles (1981) has been replaced by deep robust estimators, offering improvements in both speed and accuracy. GeoTransformer Qin et al. (2023) further enhances inlier ratios by leveraging global context with Transformers and proposing a local-to-global registration strategy that eliminates the reliance on RANSAC.

**Inter-modality Registration.** Cross-modal registration is inherently more challenging than intra-modal registration due to significant domain gaps. Early approaches typically followed a detect-then-match paradigm. For instance, 2D3D-MatchNet Feng et al. (2019) detects SIFT Ng & Henikoff (2003) and ISS Sontag (1998) keypoints and encodes local patches with CNNs and PointNet Qi et al. (2017), while P2-Net Wang et al. (2021) jointly learns keypoints and descriptors under contrastive supervision. However, keypoint-based methods are generally inefficient and less accurate, motivating detector-free strategies. For instance, 2D3D-MATR Li et al. (2023) employs a coarse-to-fine Transformer framework that establishes patch-level correspondences, refines them to fine-grained matches, and estimates rigid transformations via PnP+RANSAC Lepetit et al. (2009); Fischler & Bolles (1981). By eliminating keypoint detection and leveraging Transformers' global receptive fields, it achieves more consistent descriptors and higher inlier ratios. B2-3Dnet Cheng et al. (2025a) further enhance cross-modal correspondence learning by leveraging covariance-guided feature alignment to improve the robustness and consistency of descriptors. Based on this, CA-I2P Cheng et al. (2025b) introduces channel adaptation and global optimal selection to better align cross-modal features and reduce redundant matches, achieving improved registration accuracy. Based on 2D3D-MATR, we propose a novel framework that introduces reinforcement learning-driven overlap detection and hierarchical distribution alignment to address the limitations of existing indoor image-to-point cloud registration methods.

## 3 METHOD

Let $\mathbf{I} \in \mathbb{R}^{H \times W \times 3}$ and $\mathbf{P} \in \mathbb{R}^{N \times 3}$ be an image and a point cloud of the same scene, where $H$ and $W$ denote the image height and width, and $N$ denotes the number of points. The goal of image-to-point cloud registration is to estimate a rigid transformation $[R|t]$ in the point cloud space from the image points, where $R \in SO(3)$ is a rotation matrix and $t \in \mathbb{R}^3$ is a translation vector.

Our method adopts a detection-free paradigm, where we first extracts features and potential latent geometries of images and point clouds in overlapping regions, and the Reinforcement Learning Overlap Detector adaptively selects them through reinforcement learning strategies. Then, the Hierarchical Domain Adaptation Interaction Module alleviates the distribution shifts caused by scale inconsistencies in overlapping regions by improving multi-level cross-modal interactions between

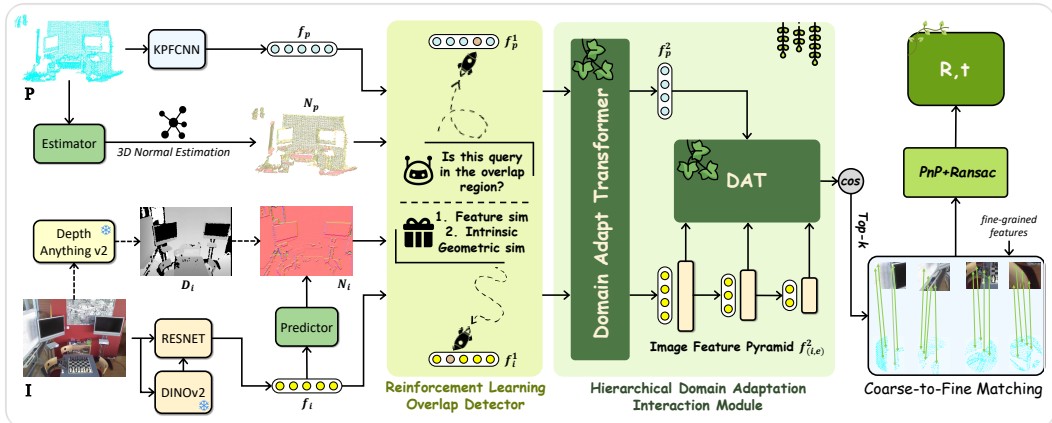

Figure 2: Overall pipeline of our method. It includes the Reinforcement Learning Overlap Detector (RLOD) and Hierarchical Domain Adaptation Interaction (HDAI) modules. In RLOD, intrinsic geometric invariants are extracted from surface normals derived from depth maps and 3D data to strengthen cross-modal correlations, while a reinforcement learning strategy adaptively selects overlapping regions for more accurate and efficient registration. In HDAI, image and point cloud features are aligned at mean and covariance levels, mitigating distribution gaps, stabilizing attention, and improving the robustness of cross-modal fusion. After achieving coarse-level matching and refining fine-level matches, PnP+RANSAC is used to regress the rigid transformation.

image and point cloud features. By computing residual similarity and top-$k$ selection, coarse correspondences are obtained, and then refined into dense correspondences using high-resolution image and point cloud features. Finally, a PnP+RANSAC Lepetit et al. (2009); Fischler & Bolles (1981) solver is applied to robustly recover the rigid transformation.

## 3.1 REINFORCEMENT LEARNING OVERLAP DETECTOR

We adopt a ResNet He et al. (2016) with FPN Lin et al. (2017) to extract image features fused with DINOv2 Oquab et al. (2023), and employ KPFCNN Thomas et al. (2019) to extract point cloud features, followed by positional encoding. The 2D and 3D features are downsampled at the lowest resolution as $f_i \in \mathbb{R}^{h \times w \times c}$ and $f_p \in \mathbb{R}^{n \times c}$, respectively. We aim to introduce 3D geometric information by leveraging surface normals, which are invariant to translation and scale, to strengthen the connection between images and point clouds and better distinguish overlapping regions, thereby reducing incorrect correspondences.

For the point cloud, since it naturally possesses 3D structure, we estimate surface normals directly from local neighborhoods: for each point, its $k$ nearest neighbors are searched, the covariance matrix of the neighborhood is computed, and the eigenvector corresponding to the smallest eigenvalue is taken as the normal of that point, followed by normalization. For the image, due to the lack of explicit 3D structure, we first employ Depth Anything v2 to predict a depth map, and then approximate depth gradients to obtain local surface orientations, from which the image normals are constructed. During training, the image branch predicts normals using a lightweight MLP (predictor), whose outputs are supervised to align with the pseudo ground-truth normals computed from the depth map, thereby ensuring geometric consistency while effectively reducing computational.

**Algorithm 1:** Normal computation and constraint for image and point cloud

**Input:** Point cloud $\mathbf{P}$, Image $\mathbf{I}$
**Output:** $\mathbf{N}_p$, $\mathbf{N}_i$, Loss $\mathcal{L}_N$
**Point cloud normals:**
**for** *each point* $p_a \in \mathbf{P}$ **do**
$\quad \mathcal{N}_a \leftarrow k\text{-NN}(p_a)$;
$\quad \mathbf{C}_a \leftarrow \frac{1}{k} \sum (p_b - \bar{p}_a)(p_b - \bar{p}_a)^\top$;
$\quad \mathbf{N}_p(a) \leftarrow \text{normalize}(\text{eigenvector}_{\min}(\mathbf{C}_a))$;

**Image normals:**
$D_i \leftarrow \text{DepthAnything}(\mathbf{I})$;
**for** *each pixel* $(u, v)$ **do**
$\quad \partial D/\partial u, \partial D/\partial v \leftarrow \text{finite diff.}$;
$\quad \mathbf{N}'_i(u, v) \leftarrow$
$\quad \quad \text{normalize}((-\partial D/\partial u, -\partial D/\partial v, 1))$;
$\quad \mathbf{N}_i(u, v) \leftarrow \text{MLP}(f_i(u, v))$;

**Normal supervision:**
$\mathcal{L}_n \leftarrow 1 - \frac{1}{hw} \sum (\mathbf{N}_i \cdot \mathbf{N}'_i)$;

However, since the surface normals of images and point clouds are computed in their respective coordinate systems, we aim to capture intrinsic geometric cues that are shared across both modalities. By characterizing the local geometric structures of image and point cloud patches, these cues provide

a reliable basis for detecting overlapping regions, thereby enabling more accurate correspondence estimation and improving the overall registration accuracy.

**Feature Sim.** For each observation unit $o$ (i.e., a corresponding image–point cloud patch), we apply average pooling (avg-pooling) over the feature vectors within the patch, obtaining the pooled image feature $f_{img}^{pool}(o)$ and the pooled point cloud feature $f_{pcd}^{pool}(o)$. Based on this, we employ cosine similarity to measure their consistency in the high-dimensional feature space:

$$\cos_f(o) = \frac{f_{img}^{pool}(o) \cdot f_{pcd}^{pool}(o)}{\|f_{img}^{pool}(o)\| \, \|f_{pcd}^{pool}(o)\|}. \tag{1}$$

Here, $o$ denotes an image–point cloud patch pair, and the metric quantifies the feature-level similarity between cross-modal patches. In addition, we directly extract matching scores from the initial image–point cloud similarity matrix as the prior value for each candidate pair:

$$S_{xy} = f_x \cdot f_y, \tag{2}$$

and select the scores of the top-$k$ candidate pairs as priors. These priors can be regarded as a weak supervision signal, indicating the reliability of candidates in the initial feature space. Such priors help guide the subsequent matching process and alleviate purely random selections to a certain extent.

**Intrinsic Geometric Sim.** To characterize the local geometric structures of image and point cloud patches, we first construct a covariance matrix from the set of normals $\{n_x \mid n_x \in N_i, \ x = 1, \ldots, X\}$, where $N_i$ denotes the previously obtained image normals and $x$ indexes the normals within a patch:

$$C_n = \frac{1}{X} \sum_{x=1}^{X} (n_x - \bar{n})(n_x - \bar{n})^T. \tag{1}$$

Then we perform eigen-decomposition to obtain eigenvalues $\lambda_1 \geq \lambda_2 \geq \lambda_3$. These eigenvalues reflect the shape of the normal distribution: if $\lambda_1 \gg \lambda_2 \approx \lambda_3$, it indicates a linear structure; if $\lambda_1 \approx \lambda_2 \gg \lambda_3$, it corresponds to a planar structure; and if the three values are approximately equal, the region is spherical or noisy. Thus, each image and point cloud patch is represented by $[\lambda_1, \lambda_2, \lambda_3]$, together forming a 6-dimensional distribution feature. To further measure the consistency of two patches in terms of overall geometric type, we normalize the eigenvalue vectors and compute the cosine similarity:

$$\tilde{\lambda} = \frac{\lambda}{\|\lambda\|}, \quad \sim_{odm} = \frac{\tilde{\lambda}_{img} \cdot \tilde{\lambda}_{pcd}}{\|\tilde{\lambda}_{img}\| \|\tilde{\lambda}_{pcd}\|}. \tag{3}$$

This metric exhibits rotation invariance, making it suitable for evaluating the consistency of image and point cloud pairs observed from different viewpoints. In addition, we compute pairwise normal angles within each patch,

$$\theta_{xy} = \arccos(n_x \cdot n_y), \tag{4}$$

and accumulate them into a histogram $H$. The distribution similarity between image and point cloud patches is defined as:

$$\sim_{hist} = 1 - \mathrm{JS}(H_{img}, H_{pcd}), \tag{3}$$

where JS denotes the Jensen–Shannon divergence. This metric captures fine-grained consistency of normal distributions, complementing $\sim_{odm}$. Finally, we obtain 6-dimensional normal features together with the two similarity measures, which jointly form a cross-modal local geometric descriptor.

**Adaptive selection.** Because the detection of overlapping regions exhibits strong uncertainty and variability across different scenes, we formulate the candidate matching between image–point cloud pairs as a reinforcement learning (RL) process to achieve adaptive query selection. For any candidate pair $(x, y)$, its state vector is defined as

$$s_{xy} = [\lambda_{img}, \lambda_{pcd}, sim_{odm}, sim_{hist}, cos_f, prior], \tag{5}$$

where $\lambda^{img}$ and $\lambda^{pcd}$ denote the eigenvalue-based normal distribution features of image and point cloud patches, $sim_{odm}$ and $sim_{hist}$ measure rotation-invariant distribution similarity, $cos_f$ denotes the cosine similarity between pooled features, and $prior$ represents the $S_{xy}$ prior confidence.

The policy network $\pi_\theta(a|s_{xy})$ takes the state vector as input and outputs a binary decision $a \in \{0, 1\}$, where $a = 1$ indicates selecting the query and $a = 0$ indicates discarding it:

$$p_{xy} = \sigma(\text{MLP}_\theta(s_{xy})), \quad \pi_\theta(a|s_{xy}) = \text{Bernoulli}(p_{xy}). \tag{6}$$

In the unsupervised setting, the reward is defined as a weighted combination of all geometric and appearance similarity components:

$$R = \beta_1 \cdot sim_\lambda + \beta_2 \cdot sim_{odm} + \beta_3 \cdot sim_{hist} + \beta_4 \cdot cos_f, \tag{7}$$

where $sim_\lambda$ denotes the cos similarity between $\lambda_{img}$ and $\lambda_{pcd}$, $\beta_k$ are balancing coefficients controlling the contribution of each term. To reduce variance in gradient estimation, we adopt an exponentially moving average baseline:

$$b \leftarrow (1 - \alpha_b)b + \alpha_b \, \mathbb{E}[R], \tag{8}$$

where $\alpha_b$ is the update rate. The final optimization objective is a regularized reinferce loss:

$$L_r = -\mathbb{E}_{a \sim \pi_\theta(s)} \big[ (R - b) \cdot \log \pi_\theta(a|s) \big] - \gamma H(\pi_\theta), \tag{9}$$

where $\gamma$ controls the exploration strength and $H(\pi_\theta)$ denotes the entropy of the Bernoulli distribution:

$$H(\pi_\theta) = -[p_{xy} \log p_{xy} + (1 - p_{xy}) \log(1 - p_{xy})]. \tag{10}$$

Through this process, the policy network learns to select candidates that are geometrically consistent across modalities. During inference, the model directly computes selection probabilities and adopts a top-$k$ Qin et al. (2022c) rule to determine the final set of query correspondences.

### 3.2 HIERARCHICAL DOMAIN ADAPTATION INTERACTION MODULE

Although RLOD enables adaptive identification of overlapping regions between images and point clouds, cross-modal registration still faces the challenge that image and point cloud features often exhibit significant distributional and scale discrepancies, which may hinder robust correspondence learning. Such discrepancies are amplified by the standard transformer Vaswani et al. (2017) attention, causing attention drift in overlaps and weakening matches. To address this, we propose the Hierarchical Domain Adaptation Interaction (HDAI) module. After the initial interaction via a Domain Adaptation Transformer (DAT), scale differences from viewpoint changes and object-size variations may still bias similarity, leading to incorrect correspondences. To mitigate this, we introduce an Image Feature Pyramid $f_{(i,n)}^2$, extracted by a lightweight CNN Zhang et al. (2022), to provide multi-scale image features. The point cloud features are then interacted with these scales to generate multiple cross-modal cosine similarity maps.

**Domain Adaptation Transformer.** DAT differs from the standard transformer by explicitly introducing a statistical distribution alignment constraint in the cross-attention key space, aiming to reduce modality discrepancy. At each layer, we compute the mean and covariance of image and point cloud features, and enforce their alignment with the following loss:

$$\begin{aligned}
\mathcal{L}_d = {} & \|\mu(f_i^1) - \mu(f_p^1)\|_2^2 + \|\Sigma(f_i^1) - \Sigma(f_p^1)\|_F^2 \\
& + \sum_{e=1}^{3} \Big( \|\mu(f_{(i,e)}^2) - \mu(f_p^2)\|_2^2 + \|\Sigma(f_{(i,e)}^2) - \Sigma(f_p^2)\|_F^2 \Big),
\end{aligned} \tag{11}$$

where $\mu(\cdot)$ and $\Sigma(\cdot)$ denote the mean and covariance of the features, respectively, $\| \cdot \|_F$ is the Frobenius norm, and $e = 1, 2, 3$ indexes the image feature pyramid scales.

During training, DAT mitigates attention drift caused by inconsistent feature scales across modalities by aligning the mean and covariance of image and point cloud features, thereby ensuring stable and reliable cross-modal attention allocation. At inference, the explicit constraint is removed; however, the learned distributional consistency is embedded in the model parameters, enabling robust and consistent cross-modal representations. Combined with multi-scale feature fusion, DAT enhances both the robustness and accuracy of registration. Finally, a top-$k$ selection preserves the most reliable similarity maps as coarse correspondences, and a PnP+RANSAC Lepetit et al. (2009); Fischler & Bolles (1981) solver is then applied to robustly recover the rigid transformation.

### 3.3 Loss Function

Let us examine the loss functions for the coarse and fine-matching networks. Both $\mathcal{L}_{\text{coarse}}$ and $\mathcal{L}_{\text{fine}}$ utilize a general circle loss Sun et al. (2020); Qin et al. (2022a). For a given anchor descriptor $d_i$, the descriptors of its positive and negative pairs are represented as $\mathcal{D}_i^P$ and $\mathcal{D}_i^N$, respectively. The matching loss function is defined as follows:

$$\mathcal{L}_i = \frac{1}{\gamma} \log \left[ 1 + \sum_{d^j \in \mathcal{D}_i^P} e^{\beta_p^{i,j}(d_i^j - \Delta_P)} \cdot \sum_{d^k \in \mathcal{D}_i^N} e^{\beta_n^{i,k}(\Delta_n - d_i^k)} \right], \tag{12}$$

where $d_i^j$ is the $L_2$ feature distance, $\beta_p^{i,j} = \gamma\lambda_p^{i,j}(d_i^j - \Delta_p)$, and $\beta_n^{i,k} = \gamma\lambda_n^{i,k}(\Delta_n - d_i^k)$ are the individual weights for the positive and negative pairs, with $\lambda_p^{i,j}$ and $\lambda_n^{i,k}$ as scaling factors. Based on the above discussion, the total loss consists of three key components: the matching loss $L_i$, the constraint between prdict and estimate normal $L_n$, the reinferce loss $L_r$ and the domain adaptation loss $L_d$, calculated as:

$$L_{\text{total}} = \omega_1 L_i + \omega_2 L_n + \omega_3 L_r + \omega_4 L_d, \tag{13}$$

where $\omega_i$ are hyperparameters balancing the contribution of different loss terms.

## 4 Experiments

### 4.1 Datasets and Implementation Details

Based on the 2D3D-MATR benchmark, we conducted extensive experiments and ablation studies on two challenging benchmarks: RGB-D Scenes v2 Lai et al. (2014) and 7-Scenes Glocker et al. (2013).

**Dataset.** *RGB-D Scenes v2* consists of 14 scenes containing furniture. For each scene, we create point cloud fragments from every 25 consecutive depth frames and sample one RGB image per 25 frames. We select image-point-cloud pairs with an overlap ratio of at least 30%. Scenes 1-8 are used for training, 9-10 for validation, and 11-14 for testing, resulting in 1,748 training pairs, 236 validation pairs, and 497 testing pairs.

The *7-Scenes* is a collection of tracked RGB-D camera frames. All seven indoor scenes were recorded from a handheld Kinect RGB-D camera at 640×480 resolution. We select image-to-point-cloud pairs from each scene with at least 50% overlap, adhering to the official sequence split for training, validation, and testing. This results in 4,048 training pairs, 1,011 validation pairs, and 2,304 testing pairs.

**Implementation Details.** We use an NVIDIA Geforce RTX 3090 GPU for training. We implement our model using PyTorch 1.13.1. For the image branch, we adopt a ResNet-50 with FPN fused with DINOv2 embeddings, and set the backbone output dimension to 128. For the point cloud branch, we employ KPFCNN with 4 stages, base voxel size of 0.025, and output dimension 128. We use the Adam optimizer with an initial learning rate of $1 \times 10^{-4}$ and weight decay $1 \times 10^{-5}$. The learning rate is decayed by a factor of 0.95 every 5 epochs. The output feature dimension of the decoder in the feature extractor is set to 512. We set $\beta_1 = \beta_2 = \beta_3 = \beta_4 = 1$ and $\omega_1 = \omega_2 = \omega_3 = 1$, $\omega_4 = 0.01$ for balancing the loss components. The number of transformer layers is set to 3.

**Metrics.** We evaluate the models using three standard metrics: Inlier Ratio (IR) — the percentage of pixel-to-point matches within 5 cm; Feature Matching Recall (FMR) — the proportion of image–point cloud pairs with IR > 10%; and Registration Recall (RR) — the proportion of pairs with RMSE below 10 cm.

### 4.2 Evaluations on Dataset

We compare our approach with baseline 2D3D-MATR Li et al. (2023) and other methods Choy et al. (2019); Wang et al. (2021); Huang et al. (2021b); Cheng et al. (2025a;b) on the RGB-D Scenes v2 and 7 Scenes dataset (see Table 1).

For the Inlier Ratio, our method achieves a mean of 46.0% on RGB-D Scenes v2 and 56.0% on 7-Scenes, surpassing the previous best results (35.5% and 51.6% from CA-I2P) by clear margins. These improvements demonstrate that the proposed Reinforcement Learning Overlap Detector (RLOD) effectively identifies reliable correspondences and reduces false matches. In terms of Feature Matching Recall, our method demonstrates consistently high performance on both datasets. It

Table 1: Evaluation results on RGB-D Scenes v2 and 7-Scenes. **Orange** and **Blue** numbers highlight the best; the second best are **boldfaced**; the baseline are underlined.

| Dataset | RGB-D Scenes v2 | | | | | 7-Scenes | | | | | | | |
|---|---|---|---|---|---|---|---|---|---|---|---|---|---|
| Model | Scene-11 | Scene-12 | Scene-13 | Scene-14 | Mean | Chess | Fire | Heads | Office | Pumpkin | Kitchen | Stairs | Mean |
| Mdpt(m) | 1.74 | 1.66 | 1.18 | 1.39 | 1.49 | 1.78 | 1.55 | 0.80 | 2.03 | 2.25 | 2.13 | 1.84 | 1.49 |
| *Inlier Ratio ↑* | | | | | | | | | | | | | |
| FCGF-2D3D | 6.8 | 8.5 | 11.8 | 5.4 | 8.1 | 34.2 | 32.8 | 14.8 | 26 | 23.3 | 22.5 | 6.0 | 22.8 |
| P2-Net | 9.7 | 12.8 | 17.0 | 9.3 | 12.2 | 55.2 | 46.7 | 13.0 | 36.2 | 32.0 | 32.8 | 5.8 | 31.7 |
| Predator-2D3D | 17.7 | 19.4 | 17.2 | 8.4 | 15.7 | 34.7 | 33.8 | 16.6 | 25.9 | 23.1 | 22.2 | 7.5 | 23.4 |
| 2D3D-MATR | 32.8 | 34.4 | 39.2 | 23.3 | 32.4 | 72.1 | 66.0 | 31.3 | 60.7 | 50.2 | 52.5 | 18.1 | 50.1 |
| B2-3Dnet | 36.4 | 32.7 | **43.8** | **27.4** | 35.1 | **73.8** | **66.7** | 33.1 | 61.7 | 50.8 | 52.3 | 18.1 | 50.9 |
| CA-I2P | **38.6** | **40.6** | 38.9 | 24.0 | **35.5** | 73.6 | 66.4 | **34.5** | **62.4** | **52.1** | **52.8** | **19.1** | **51.6** |
| Ours | 51.9 | 51.7 | 45.3 | 35.2 | 46.0 | 75.8 | 67.2 | 50.8 | 67.3 | 53.4 | 56.2 | 21.3 | 56.0 |
| *Feature Matching Recall ↑* | | | | | | | | | | | | | |
| FCGF-2D3D | 11.1 | 30.4 | 51.5 | 15.5 | 27.1 | **99.7** | 98.2 | 69.9 | 97.1 | 83.0 | 87.7 | 16.2 | 78.8 |
| P2-Net | 48.6 | 65.7 | 82.5 | 41.6 | 59.6 | 100.0 | 99.3 | 58.9 | **99.1** | 87.2 | 92.2 | 16.2 | 79 |
| Predator-2D3D | 86.1 | 89.2 | 63.9 | 24.3 | 65.9 | 91.3 | 95.1 | 76.6 | 88.6 | 79.2 | 80.6 | 31.1 | 77.5 |
| 2D3D-MATR | **98.6** | 98.0 | 88.7 | 77.9 | 90.8 | 100.0 | 99.6 | 98.6 | 100.0 | **92.4** | **95.9** | 58.2 | 92.1 |
| B2-3Dnet | 100.0 | **99.0** | 92.8 | 85.8 | 94.4 | 100.0 | 100.0 | 98.6 | 100.0 | 92.7 | 95.6 | **64.9** | 93.1 |
| CA-I2P | 100.0 | 100.0 | 91.8 | 82.7 | 93.6 | 100.0 | 100.0 | 98.6 | 100.0 | 92.0 | 95.5 | 60.8 | 92.4 |
| Ours | 100.0 | 100.0 | 91.8 | 87.2 | 94.7 | 100.0 | 100.0 | 100.0 | 100.0 | 90.3 | 96.3 | 65.1 | 93.1 |
| *Registration Recall ↑* | | | | | | | | | | | | | |
| FCGF-2D3D | 26.5 | 41.2 | 37.1 | 16.8 | 30.4 | 89.5 | 79.7 | 19.2 | 85.9 | 69.4 | 79.0 | 6.8 | 61.4 |
| P2-Net | 40.3 | 40.2 | 41.2 | 31.9 | 38.4 | 96.9 | 86.5 | 20.5 | 91.7 | 75.3 | 85.2 | 4.1 | 65.7 |
| Predator-2D3D | 44.4 | 41.2 | 21.6 | 13.7 | 30.2 | 69.6 | 60.7 | 17.8 | 62.9 | 56.2 | 62.6 | 9.5 | 48.5 |
| 2D3D-MATR | 63.9 | 53.9 | 58.8 | 49.1 | 56.4 | 96.9 | 90.7 | 52.1 | 95.5 | 80.9 | 86.1 | 28.4 | 75.8 |
| B2-3Dnet | 58.3 | 60.8 | **74.2** | **60.2** | 63.4 | 98.3 | 90.5 | 56.2 | **96.4** | **84.0** | 86.1 | 32.4 | 77.7 |
| CA-I2P | **68.1** | **73.5** | 63.9 | 47.8 | **63.3** | **99.0** | 90.7 | **68.5** | 96.2 | 83.0 | **88.1** | **33.1** | **79.5** |
| Ours | 91.7 | 90.2 | 86.6 | 73.5 | 85.5 | 99.7 | 96.0 | 94.5 | 98.7 | 84.0 | 92.7 | 36.5 | 86.0 |

achieves perfect recall (100.0%) in most scenes and shows clear improvements in more challenging cases, such as Scene-14 and Stairs, with gains of 4.5% and 4.3%, respectively. Compared with prior works, these results demonstrate that the proposed Hierarchical Domain Adaptation Interaction (HDAI) module effectively aligns cross-modal features and stabilizes attention computation, thereby ensuring robust matching performance. For the final metric of Registration Recall, our method achieves state-of-the-art performance on both datasets. On RGB-D Scenes v2, it reaches 85.5%, which is a clear improvement over CA-I2P (73.5%) and 2D3D-MATR (72.0%). On 7-Scenes, our approach further achieves 93.1%, surpassing CA-I2P (92.4%) and B2-3Dnet (77.7%). These results highlight the effectiveness of our overlap detection and feature alignment strategies in delivering accurate and reliable indoor registration across diverse scenes. In Appendix A.5, we present further metrics validation and comparisons with additional methods.

## 4.3 ABLATION STUDIES

We conduct ablation studies on RGB-D Scenes v2 to evaluate the contribution of each component (Table 2). Starting from the baseline, the model achieves 32.4% IR, 90.8% FMR, and 56.4% RR. When introducing the RLOD module (M1), the RR increases dramatically from 56.4% to 83.4%, while the IR also rises from 32.4% to 45.4%. This demonstrates that reinforcement learning–based overlap detection is crucial for identifying reliable correspondences and reducing mismatches. In comparison, removing intrinsic geometric cues (M2) leads to a decrease to 42.6% IR and 80.7%

Table 2: Ablation studies of modules on RGB-D Scenes v2.

| Exp. | RLOD | HDAI | Intrinsic Geometric | Hierarchical DAT | DINO | IR↑ | FMR↑ | RR↑ |
|---|---|---|---|---|---|---|---|---|
| Base | ✗ | ✗ | ✗ | ✗ | ✗ | 32.4 | 90.8 | 56.4 |
| M1 | ✓ | ✗ | ✓ | ✗ | ✓ | 45.4 | 93.6 | 83.4 |
| M2 | ✓ | ✗ | ✗ | ✗ | ✓ | 42.6 | 93.2 | 80.7 |
| M3 | ✗ | ✓ | ✗ | ✓ | ✓ | 42.5 | 92.8 | 78.2 |
| M4 | ✗ | ✓ | ✗ | ✗ | ✓ | 41.9 | 92.3 | 76.5 |
| M6 | ✗ | ✗ | ✗ | ✗ | ✓ | 37.8 | 92.4 | 73.2 |
| Full | ✓ | ✓ | ✓ | ✓ | ✓ | **46.0** | **94.7** | **85.5** |

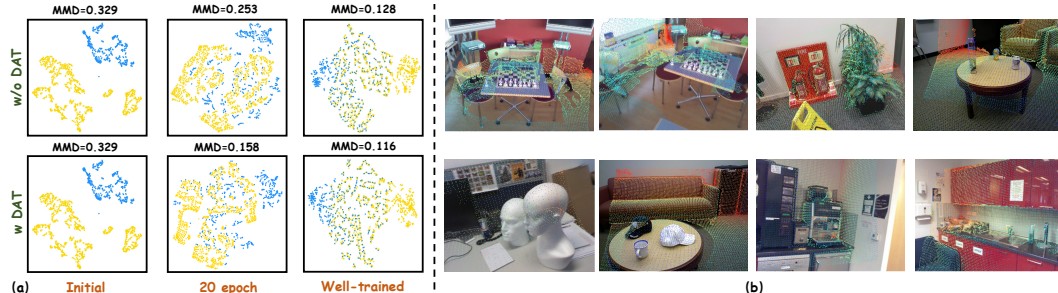

Figure 3: (a) t-SNE visualizations with and without DAT after 20 epochs and at the end of training, along with corresponding MMD values. (b) Visualization of point cloud projections onto the image.

RR, showing that relying solely on surface normals for implicit modeling provides limited benefits. Adding the HDAI module (M3) improves the results to 42.5% IR and 78.2% RR, whereas removing hierarchical domain adaptation (M4) causes performance to drop to 41.9% IR and 76.5% RR due to scale mismatch. These results confirm that aligning feature distributions at multiple statistical levels effectively stabilizes cross-modal attention. Experiment M6, which keeps only DINO features, achieves 37.8% IR and 73.2% RR. Finally, the full model that integrates all modules attains the best performance with 46.0% IR, 94.7% FMR, and 85.5% RR, demonstrating the complementary benefits of RLOD, geometric cues, and HDAI for robust indoor registration. Appendix A.6 includes more ablation studies and a detailed comparison of computational complexity.

### 4.4 QUANTITATIVE ANALYSIS

Although extensive experiments have been conducted, we also perform comprehensive visualization analyses to intuitively demonstrate the model's performance. In Figure 3, (a) presents t-SNE visualizations and corresponding MMD values (see Appendix A.1) under different conditions. DAT accelerates domain discrepancy reduction and achieves better alignment between images and point clouds after training, facilitating accurate registration. In (b), we project the point clouds onto the images using the estimated poses, showing no large-scale misalignment across scenes, indicating satisfactory registration.

In Figure 4, we use a 40-pixel projection distance threshold to classify matches as correct (green lines) or incorrect (red lines). The RLOD module effectively identifies overlapping regions, reducing misalignments, while the HDAI module alleviates domain discrepancies, improving registration performance. Appendix A.7 provides additional visual results, and Appendix A.8 offers further insights into these visualizations.

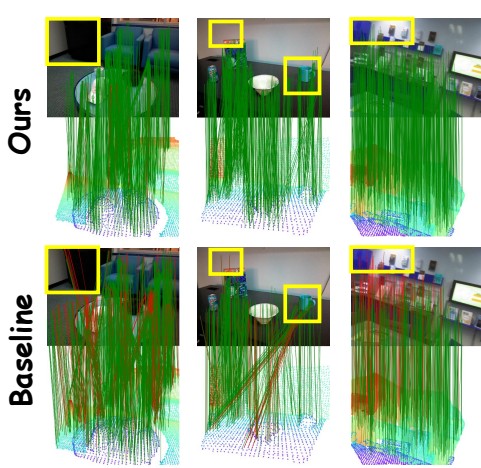

Figure 4: Qualitative registration results with projection-based matching visualization.

## 5 CONCLUSION

In this paper, we presented a novel framework for Learning Overlap Detection in Domain-Adaptive Image-to-Point Cloud Registration. Our approach introduces two key components: the Reinforcement Learning Overlap Detector (RLOD), which leverages intrinsic geometric cues from surface normals to enrich image representations and adaptively identify overlapping regions, and the Hierarchical Domain Adaptation Interaction (HDAI) module, which aligns cross-modal feature distributions at multiple statistical levels to stabilize attention and mitigate scale mismatches. Through the complementary effects of reliable overlap detection and robust feature alignment, our method achieves substantial improvements in both accuracy and robustness for indoor registration. Extensive experiments on RGB-D Scenes v2 and 7-Scenes validate the effectiveness of our framework, establishing new state-of-the-art performance in image-to-point cloud registration.

## ETHICS STATEMENT

This research adheres to the ICLR Code of Ethics. We ensure that no ethical violations have occurred during the research process. All datasets used comply with publicly available privacy policies, and we have ensured the security and privacy of the data during collection and use. There are no conflicts of interest or funding issues in this research. All methods and applications used in this research follow principles of fairness and objectivity to ensure the integrity and transparency of the research.

## REPRODUCIBILITY STATEMENT

All improvements in this research are based on open-source code and datasets. We provide comprehensive experimental details and algorithm descriptions, including the models, datasets, and training processes used. All relevant source code and datasets will be made open-source. We encourage readers to use the same experimental setups and parameters to reproduce our results and validate the theories and algorithms presented in this work, ensuring the reproducibility of the research and supporting the validation of the results.

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

# A APPENDIX

## A.1 T-SNE AND MMD

**t-SNE.** t-Distributed Stochastic Neighbor Embedding (t-SNE) Van der Maaten & Hinton (2008) is a nonlinear dimensionality reduction technique used primarily for data visualization. It is particularly effective for visualizing high-dimensional datasets by embedding them into two or three dimensions. t-SNE aims to preserve the local structure of data points by modeling similar objects with nearby points and dissimilar objects with distant points. This method is beneficial for visualizing multi-modal tasks, allowing for intuitive insights into complex datasets that span various domains such as images, text, and audio.

**MMD.** Maximum Mean Discrepancy (MMD) is a statistical method used to compare two probability distributions. It is a non-parametric technique that measures the difference between distributions by mapping data into a high-dimensional feature space using a kernel function. MMD is widely used in various machine learning applications, such as generative adversarial networks (GANs) Salimans et al. (2016), domain adaptation, and distribution testing. The core idea of MMD is to compute the distance between the means of two distributions in a reproducing kernel Hilbert space (RKHS) Berlinet & Thomas-Agnan (2011). Given two distributions $I$ and $Q$, the MMD is defined as:

$$\text{MMD}(I, Q) = \left\| \mathbb{E}_{x \sim I}[\phi(x)] - \mathbb{E}_{y \sim Q}[\phi(y)] \right\|_{\mathcal{H}}, \tag{14}$$

where $\phi$ is the feature mapping function induced by a kernel, and $\mathcal{H}$ is the RKHS.

MMD is applied in various areas, including generative models for evaluating the similarity between the distributions of generated and real data, domain adaptation for reducing distribution shifts between source and target domains, and hypothesis testing for determining if two samples are drawn from the same distribution.

## A.2 POSITIONAL EMBEDDING

We augment the 2D and 3D features with their positional information before the attention layer.

$$\hat{F}^{\mathcal{I}}_{\text{pos}} = \hat{F}^{\mathcal{I}} + \phi(\hat{Q}), \quad \hat{F}^{\mathcal{P}}_{\text{pos}} = \hat{F}^{\mathcal{P}} + \phi(\hat{P}). \tag{15}$$

The Fourier embedding function $\phi(x)$ Mildenhall et al. (2021) encodes positional information by transforming it into a sequence of sine and cosine terms:

$$\phi(x) = \left[ x, \sin(2^0 x), \cos(2^0 x), \ldots, \sin(2^{L-1} x), \cos(2^{L-1} x) \right], \tag{16}$$

where $L$ is the length of the embedding. This transformation incorporates spatial positioning into the features. To facilitate further computations, the first two spatial dimensions of the 2D features are flattened, making the augmented features $\hat{F}_{\text{pos}}^{\mathcal{I}}$ and $\hat{F}_{\text{pos}}^{\mathcal{P}}$ ready for subsequent processing.

### A.3 NETWORK ARCHITECTURE

We utilize a 4-stage ResNet He et al. (2016) with a Feature Pyramid Network (FPN) Lin et al. (2017) as the image backbone. The output channels for each stage are $\{128, 128, 256, 512\}$. The input images have a resolution of $480 \times 640$ pixels, which is downsampled to $60 \times 80$ in the coarsest level for efficiency. For the 3D backbone, a 4-stage KPFCNN Thomas et al. (2019) is employed, with output channels configured as $\{128, 256, 512, 1024\}$. Point clouds are voxelized with an initial voxel size of 2.5 cm, which doubles at each stage.

At the coarse level, 2D features are resized to $24 \times 32$ pixels before being fed into the transformer to enhance computational efficiency. Each transformer layer has 256 feature channels, 4 attention heads, and uses ReLU activation functions. In the patch pyramid setup, the coarsest level begins with $H_0 = 6$ and $W_0 = 8$, expanding through 3 pyramid levels: $\{6 \times 8, 12 \times 16, 24 \times 32\}$. At the fine level, both 2D and 3D features are projected into a 128-dimensional space for feature matching.

To address significant misalignments caused by structurally similar but non-overlapping regions, we incorporate ground-truth supervision during training by evaluating the overlap ratio between the annotated image-point cloud correspondences, using the dataset-provided ground-truth pose. This overlap measure is computed based on the local neighborhoods of image keypoints and point cloud nodes, which are projected according to the ground-truth pose. However, relying solely on this overlap criterion results in sparse and discontinuous reward signals, as only a limited number of correspondences exhibit valid overlaps, leaving most candidate pairs unlabeled. We incorporate the overlap signal in combination with the rotation-invariant geometric similarity to form the final reward signal for policy optimization. This addition helps prevent large misalignments by providing supervisory information when available, without significantly influencing the reinforcement learning process in regions lacking explicit annotations. As such, it ensures the network benefits from strong supervision in well-annotated areas, while still receiving dense and consistent feedback in less annotated regions, thereby supporting stable learning.

We define ground truth using bilateral overlap Huang et al. (2021a). A patch pair is positive if both the 2D and 3D overlap ratios are at least 30%, and negative if both are below 20%. The 2D and 3D overlap ratios are used as $\lambda_p$, while $\lambda_n$ is set to 1. At the fine level, a pixel-point pair is positive if the 3D distance is below 3.75 cm and the 2D distance is under 8 pixels, and negative if the 3D distance exceeds 10 cm or the 2D distance is over 12 pixels. Scaling factors are set to 1. Pairs not meeting these criteria are ignored as the safe region during training. The margins are set to $\Delta_p = 0.1$ and $\Delta_n = 1.4$.

### A.4 METRICS

#### A.4.1 COMMON METRICS

We evaluate our method using several key metrics: Inlier Ratio (IR), Feature Matching Recall (FMR), and Registration Recall (RR).

**Inlier Ratio** (IR) quantifies the proportion of inliers among all putative pixel-point correspondences. A correspondence is deemed an inlier if its 3D distance is less than a threshold $\tau_1 = 5$ cm under the ground-truth transformation $\mathbf{T}_{\mathcal{P} \to \mathcal{I}}^*$:

$$\text{IR} = \frac{1}{|C|} \sum_{(x_i, y_i) \in C} \left[ \left\| \mathbf{T}_{\mathcal{P} \to \mathcal{I}}^*(x_i) - \mathbf{K}^{-1}(y_i) \right\|_2 < \tau_1 \right], \tag{17}$$

Here, $[\cdot]$ denotes the Iverson bracket, $x_i \in \mathcal{P}$, and $y_i \in \mathcal{Q} \subseteq \mathcal{I}$ are pixel coordinates. The function $\mathbf{K}^{-1}$ projects a pixel to a 3D point based on its depth value.

**Feature Matching Recall** (FMR) represents the fraction of image-point-cloud pairs with an IR above a threshold $\tau_2 = 0.1$. It measures the likelihood of successful registration:

$$\text{FMR} = \frac{1}{M} \sum_{i=1}^{M} \left[ \text{IR}_i > \tau_2 \right], \tag{18}$$

where $M$ is the total number of image-point-cloud pairs.

**Registration Recall** (RR) measures the fraction of image-point-cloud pairs that are correctly registered. A pair is correctly registered if the root mean square error (RMSE) between the ground-truth-transformed and predicted point clouds $\mathbf{T}_{\mathcal{P} \to \mathcal{I}}$ is less than $\tau_3 = 0.1$ m:

$$\text{RMSE} = \sqrt{\frac{1}{|\mathcal{P}|} \sum_{p_i \in \mathcal{P}} \|\mathbf{T}_{\mathcal{P} \to \mathcal{I}}(p_i) - \mathbf{T}^*_{\mathcal{P} \to \mathcal{I}}(p_i)\|_2^2}, \tag{19}$$

$$\text{RR} = \frac{1}{M} \sum_{i=1}^{M} \left[ \text{RMSE}_i < \tau_3 \right], \tag{20}$$

These metrics provide a comprehensive evaluation of the model's capability to accurately match features and register image-point-cloud pairs, ensuring robust alignment and effective correspondence.

### A.4.2 OTHER METRICS

There are also some other metrics that can be used to test the model's performance, and we conducted tests using those as well.

**Patch Inlier Ratio** (PIR). we introduced PIR Qin et al. (2022b) as an additional metric, measuring the ratio of patch correspondences with overlap ratios greater than 0.3. This metric assesses performance at the coarse level. PIR is a crucial metric for assessing the performance of patch correspondence algorithms. It measures the fraction of patch correspondences whose overlap ratios exceed a certain threshold, typically set at 0.3. Specifically, PIR reflects the quality of the putative patch correspondences under the ground-truth transformation. A pixel (or point) is considered overlapped if its 3D distance is below a specified threshold (e.g., 3.75 cm) and its 2D distance is below another threshold (e.g., 8 pixels). For each patch, we calculate two overlap ratios—one on the image side and one on the point cloud side—and take the smaller as the final overlap ratio. In our ablation study, we utilized these metrics to evaluate the performance.

**Relative Rotation Error** (RRE) is a metric used to evaluate the accuracy of estimated rotations in 3D space. It calculates the total angular error between the estimated rotation and the ground-truth rotation by summing the absolute values of the Euler angles representing the rotation difference:

$$\text{RRE} = \sum_{i=1}^{3} |r(i)|, \tag{21}$$

where $r$ is the Euler angle vector obtained from the product of the inverse of the ground-truth rotation matrix $R_{\text{gt}}$ and the estimated rotation matrix $R_E$.

**Relative Translation Error** (RTE) measures the discrepancy between the estimated and ground-truth translation vectors in 3D space. It is defined as the Euclidean distance between the two vectors, quantifying the translation accuracy:

$$\text{RTE} = \|\mathbf{t}_{\text{gt}} - \mathbf{t}_E\|_2, \tag{22}$$

where $\mathbf{t}_{\text{gt}}$ is the ground-truth translation vector and $\mathbf{t}_E$ is the estimated translation vector. The Euclidean norm $\|\cdot\|_2$ computes the straight-line distance between these vectors.

### A.5 ADDITIONAL EXPERIMENTS

As a registration task, prior methods have often overlooked rotation-related metrics. To address this, we conduct a comprehensive comparison with the baseline and previous state-of-the-art using four

Table 3: Evaluation results on *RGB-D Scenes v2* and *7-Scenes* using Relative Rotation Error (RRE) and Relative Translation Error (RTE). **Bold-faced** numbers highlight the best and the second best are underlined.

| Dataset | *RGB-D Scenes v2* | | | | | *7-Scenes* | | | | | | | |
|---|---|---|---|---|---|---|---|---|---|---|---|---|---|
| Model | Scene-11 | Scene-12 | Scene-13 | Scene-14 | Mean | Chess | Fire | Heads | Office | Pumpkin | Kitchen | Stairs | Mean |
| Mdpt(m) | 1.74 | 1.66 | 1.18 | 1.39 | 1.49 | 1.78 | 1.55 | 0.80 | 2.03 | 2.25 | 2.13 | 1.84 | 1.49 |
| *Mean_RRE*(°) ↓ | | | | | | | | | | | | | |
| 2D3D-MATR | 2.294 | 2.628 | 3.823 | 3.358 | 3.026 | 2.298 | 3.144 | 7.549 | 2.285 | 2.439 | 2.620 | 2.705 | 3.291 |
| B2-3Dnet | 2.245 | 2.317 | 2.604 | 3.233 | 2.600 | 2.205 | 3.105 | 7.414 | 2.294 | 2.421 | 2.618 | 2.564 | 3.232 |
| CA-I2P | 2.008 | 2.031 | 3.306 | 2.890 | 2.559 | 2.354 | 3.093 | 7.105 | 2.272 | 2.297 | 2.624 | 2.654 | 3.200 |
| Ours | 1.120 | 1.639 | 2.440 | 2.455 | 1.913 | 1.885 | 2.759 | 5.526 | 1.971 | 2.262 | 2.399 | 2.563 | 2.766 |
| *Mean_RTE*(m) ↓ | | | | | | | | | | | | | |
| 2D3D-MATR | 0.066 | 0.086 | 0.067 | 0.088 | 0.077 | 0.054 | 0.084 | 0.088 | 0.069 | 0.088 | 0.076 | 0.090 | 0.079 |
| B2-3Dnet | 0.056 | 0.061 | 0.041 | 0.079 | 0.059 | 0.051 | 0.082 | 0.093 | 0.066 | 0.083 | 0.074 | 0.086 | 0.076 |
| CA-I2P | 0.057 | 0.061 | 0.054 | 0.072 | 0.061 | 0.054 | 0.082 | 0.094 | 0.069 | 0.078 | 0.077 | 0.085 | 0.076 |
| Ours | 0.043 | 0.049 | 0.042 | 0.067 | 0.050 | 0.039 | 0.072 | 0.067 | 0.056 | 0.078 | 0.065 | 0.086 | 0.066 |
| *Median_RRE*(°) ↓ | | | | | | | | | | | | | |
| 2D3D-MATR | 1.995 | 2.335 | 3.074 | 3.194 | 2.649 | 1.972 | 2.821 | 6.903 | 2.025 | 2.192 | 2.364 | 3.129 | 3.058 |
| B2-3Dnet | 2.192 | 2.227 | 2.099 | 3.121 | 2.410 | 1.942 | 2.761 | 7.163 | 2.018 | 2.277 | 2.350 | 2.408 | 2.988 |
| CA-I2P | 1.789 | 1.763 | 2.826 | 2.383 | 2.190 | 1.977 | 2.722 | 5.997 | 1.985 | 2.144 | 2.392 | 2.560 | 2.825 |
| Ours | 1.315 | 1.514 | 1.935 | 1.842 | 1.615 | 1.690 | 2.321 | 4.864 | 1.833 | 2.135 | 2.235 | 2.226 | 2.472 |
| *Median_RTE*(m) ↓ | | | | | | | | | | | | | |
| 2D3D-MATR | 0.058 | 0.072 | 0.055 | 0.080 | 0.066 | 0.047 | 0.079 | 0.082 | 0.065 | 0.084 | 0.069 | 0.105 | 0.076 |
| B2-3Dnet | 0.057 | 0.058 | 0.053 | 0.070 | 0.055 | 0.048 | 0.074 | 0.095 | 0.063 | 0.079 | 0.069 | 0.085 | 0.073 |
| CA-I2P | 0.055 | 0.054 | 0.048 | 0.065 | 0.055 | 0.048 | 0.076 | 0.078 | 0.062 | 0.071 | 0.070 | 0.079 | 0.069 |
| Ours | 0.036 | 0.045 | 0.035 | 0.056 | 0.043 | 0.035 | 0.063 | 0.064 | 0.048 | 0.073 | 0.056 | 0.069 | 0.060 |

metrics, including Relative Rotation Error (RRE) and Relative Translation Error (RTE), as shown in Table 3.

Specifically, for mean rotation error (Mean_RRE), our approach reduces the error to 1.913° on RGB-D Scenes v2 and 2.766° on 7-Scenes, outperforming the best baseline CA-I2P (2.559° / 3.200°). Similarly, for mean translation error (Mean_RTE), our method obtains 0.050 m and 0.066 m, clearly surpassing B2-3Dnet (0.059 m / 0.076 m). Consistent improvements are also observed in the median metrics, where our approach achieves 1.615° / 2.472° (Median_RRE) and 0.043 m / 0.060 m (Median_RTE), outperforming all baselines by a notable margin. We further observe that in more challenging scenes such as Scene-13, Kitchen, and Stairs, the improvements in the median RRE and RTE are even more pronounced compared to the mean values, indicating that our method not only achieves overall performance gains but is also particularly effective in complex scenarios with

Table 4: Evaluation results with FreeReg on RGB-D Scenes v2. **Purple** numbers highlight the best, the second best are **bold**, and the baseline are underlined.

| Model | Scene-11 | Scene-12 | Scene-13 | Scene-14 | Mean |
|---|---|---|---|---|---|
| Mean depth (m) | 1.74 | 1.66 | 1.18 | 1.39 | 1.49 |
| *Inlier Ratio* ↑ | | | | | |
| 2D3D-MATR | 32.8 | 34.4 | **39.2** | 23.3 | 32.4 |
| FreeReg | **36.6** | **34.5** | 34.2 | 18.2 | 30.9 |
| Ours | 51.9 | 51.7 | 45.3 | 35.2 | 46.0 |
| *Feature Matching Recall* ↑ | | | | | |
| 2D3D-MATR | **98.6** | **98.0** | 88.7 | **77.9** | **90.8** |
| FreeReg | 91.9 | 93.4 | **93.1** | 49.6 | 82.0 |
| Ours | 100.0 | 100.0 | 91.8 | 87.2 | 94.7 |
| *Registration Recall* ↑ | | | | | |
| 2D3D-MATR | 63.9 | 53.9 | **58.8** | **49.1** | 56.4 |
| FreeReg | **74.2** | **72.5** | 54.5 | 27.9 | **57.3** |
| Ours | 91.7 | 90.2 | 86.6 | 73.5 | 85.5 |

repetitive textures or feature extraction limitations. These results demonstrate the superiority of our overlap detection and feature alignment strategies in reducing both rotational and translational errors, thereby ensuring more accurate and reliable image-to-point cloud registration.

Some methods are not included in the main text due to space limitations, and we provide their comparison here. FreeReg Wang et al. (2023) is a zero-shot image-to-point cloud registration method that uses pretrained diffusion models to extract "diffusion features" and monocular depth estimators to compute geometric features, enabling robust pixel-to-point correspondences across modalities without any task-specific training. Table 4 reports the evaluation results on RGB-D Scenes v2 with baseline Li et al. (2023) and FreeReg in terms of Inlier Ratio (IR), Feature Matching Recall (FMR), and Registration Recall (RR). Compared with them, our method achieves significant improvements across all metrics. Specifically, our approach reaches an average IR of 46.0%, substantially higher than 32.4% for 2D3D-MATR and 30.9% for FreeReg, demonstrating its effectiveness in producing more reliable correspondences. For FMR, our method attains 94.7%, outperforming both 2D3D-MATR (90.8%) and FreeReg (82.0%). Most notably, the RR of our method improves to 85.5%, a large margin over 56.4% (2D3D-MATR) and 57.3% (FreeReg). These results highlight that by integrating reinforcement learning–based overlap detection and hierarchical domain adaptation, our method not only ensures higher-quality matches but also achieves substantially more robust registration compared to existing approaches.

Diff$^2$I2P Mu et al. (2025) is a new fully differentiable image-to-point cloud registration framework that leverages a depth-conditioned diffusion prior via Control-Side Score Distillation and Deformable Correspondence Tuning, significantly improving cross-modal alignment performance. Table 5 reports the detailed evaluation results on 7-Scenes, comparing our method with baseline and Diff$^2$I2P across all scenes. For Inlier Ratio (IR), our approach consistently outperforms the baselines, achieving the best average of 56.0%, with particularly large gains in challenging scenes such as Heads (50.8% vs. 39.2%) and Stairs (21.3% vs. 18.1%). In terms of Feature Matching Recall (FMR), our method reaches 93.1% on average, surpassing both 2D3D-MATR (92.1%) and Diff$^2$I2P (92.2%), and notably improving the matching robustness in the Stairs scene (65.1% vs. 55.4%). For the most important metric, Registration Recall (RR), our model achieves an average of 86.0%, setting a new state of the art. The improvements are particularly significant in Heads (94.5% vs. 74.0%) and Kitchen (92.7% vs. 90.2%), demonstrating the effectiveness of our overlap detection and feature alignment strategies in handling indoor scenes with repetitive textures and structural ambiguities.

Table 6 compares the performance of our method with two baseline approaches: 2D3D-MATR and Diff$^2$I2P. Our method achieves significant improvements across all metrics. Specifically, we achieve 74.8% PIR, 46.2% IR, 94.7% FMR, and 85.5% RR, surpassing both 2D3D-MATR and Diff$^2$I2P

Table 5: Evaluation results with Diff$^2$I2P on 7-Scenes. **Purple** numbers highlight the best, the second best are **Boldfaced**, and the baseline are underlined.

| Model | Chess | Fire | Heads | Office | Pumpkin | Kitchen | Stairs | Mean |
|---|---|---|---|---|---|---|---|---|
| Mean depth (m) | 1.78 | 1.55 | 0.80 | 2.03 | 2.25 | 2.13 | 1.84 | 1.49 |
| *Inlier Ratio* ↑ | | | | | | | | |
| 2D3D-MATR | 72.1 | 66.0 | 31.3 | 60.7 | 50.2 | 52.5 | **18.1** | 50.1 |
| Diff$^2$I2P | **74.1** | 68.8 | 39.2 | 65.6 | 52.1 | 54.2 | 18.1 | 53.2 |
| Ours | 75.8 | 67.2 | 50.8 | 67.3 | 53.4 | 56.2 | 21.3 | 56.0 |
| *Feature Matching Recall* ↑ | | | | | | | | |
| 2D3D-MATR | 100.0 | 99.6 | 98.6 | 100.0 | **92.4** | 95.9 | **58.2** | 92.1 |
| Diff$^2$I2P | 100.0 | 100.0 | 100.0 | 100.0 | 93.4 | 96.2 | 55.4 | 92.2 |
| Ours | 100.0 | 100.0 | 100.0 | 100.0 | 90.3 | 96.3 | 65.1 | 93.1 |
| *Registration Recall* ↑ | | | | | | | | |
| 2D3D-MATR | 96.9 | 90.7 | 52.1 | 95.5 | 80.9 | 86.1 | **28.4** | 75.8 |
| Diff$^2$I2P | 99.0 | 95.6 | 74.0 | 98.9 | 86.8 | 90.2 | 36.5 | 83.0 |
| Ours | 99.7 | 96.0 | 94.5 | 98.7 | 84.0 | 92.7 | 36.5 | 86.0 |

by a notable margin. These results demonstrate the effectiveness of our approach in enhancing the reliability of correspondences and improving overall registration performance.

Table 6: Evaluation results with Diff$^2$I2P on RGB-D Scenes v2

| Method | PIR | IR | FMR | RR |
|---|---|---|---|---|
| 2D3D-MATR | 57.6 | 32.4 | 90.8 | 56.4 |
| Diff$^2$I2P | 60.8 | 36.9 | 77.1 | 60.5 |
| Ours | **74.8** | **46.2** | **94.7** | **85.5** |

## A.6 ADDITIONAL ABLATION STUDIES

To make our experimental results more convincing, we further conducted extensive ablation studies for verification.

Table 6 compares the efficiency of our method with MATR-2D3D. Our model requires slightly more memory (7942 MB vs. 6240 MB) due to the additional modules, and converges within 12 epochs compared to 11 epochs for MATR-2D3D.

Table 7: Comparison of memory consumption and convergence speed.

| Method | Baseline | Ours |
|---|---|---|
| Memory (MB) | 6240 | 7942 |
| Convergence | 11 epochs | 12 epochs |

Table 8: Computation time per iteration of different variants.

| Method | Baseline | w DepthAnything v2 | w DAT | Full |
|---|---|---|---|---|
| Computation time per iteration (s) | 0.147 | 0.254 | 0.173 | 0.153 |

To provide a clearer comparison of computational efficiency, we report the computation time per iteration for different variants in Table 8. The baseline model requires 0.147 s per iteration, while directly incorporating DepthAnything v2 significantly increases the time to 0.254 s due to the overhead of dense depth estimation. Retaining the DAT loss during inference results in 0.173 s per iteration, representing only a moderate increase over the baseline. Our full model runs at 0.153 s per iteration, showing that the additional modules introduce only a slight overhead through our careful design. Considering the substantial improvements in registration accuracy reported in previous experiments, this trade-off between efficiency and performance is both reasonable and well justified.

We further conduct a parameter sensitivity analysis on the weight $\omega_4$ (Table 9). The results show that our method is relatively robust to different choices of $\omega_4$, with only minor fluctuations across all metrics. Among the tested values, setting $\omega_4 = 0.01$ achieves the best overall balance, yielding the

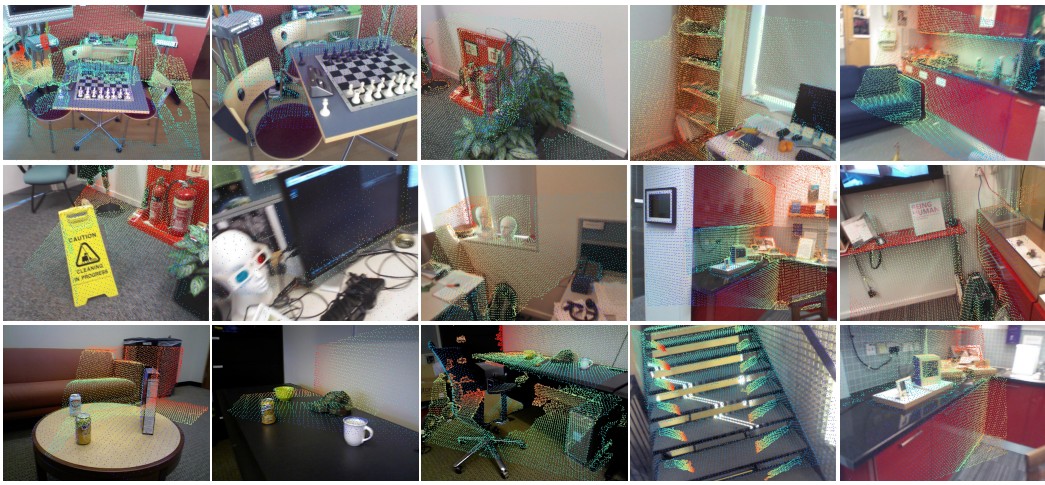

Figure 5: More projection visualization on Dataset.

Table 9: Parameter sensitivity analysis on the weight $\omega_4$.

| $\omega_4$ | 0.005 | 0.01 | 0.02 | 0.05 | 0.1 |
|---|---|---|---|---|---|
| PIR | 71.6 | **74.8** | 74.1 | 74.4 | 73.3 |
| IR | 44.6 | 46.0 | **46.2** | 45.3 | 44.5 |
| FMR | 93.9 | **94.7** | 92.5 | 92.1 | 92.8 |
| RR | 82.0 | **85.5** | 82.9 | 81.7 | 79.8 |

highest IR (46.0%), FMR (94.7%), and RR (85.5%). These results indicate that while $\omega_4$ has some effect on performance, our method maintains stable accuracy within a reasonable range of parameter choices, demonstrating its robustness. However, if $\omega_4$ is set too large, it excessively emphasizes the distribution alignment loss, which could dominate the feature interaction process, disrupting the balance between feature alignment and other model objectives. This may cause over-regularization, reducing the model's ability to adaptively refine cross-modal correspondences and impacting the overall registration accuracy.

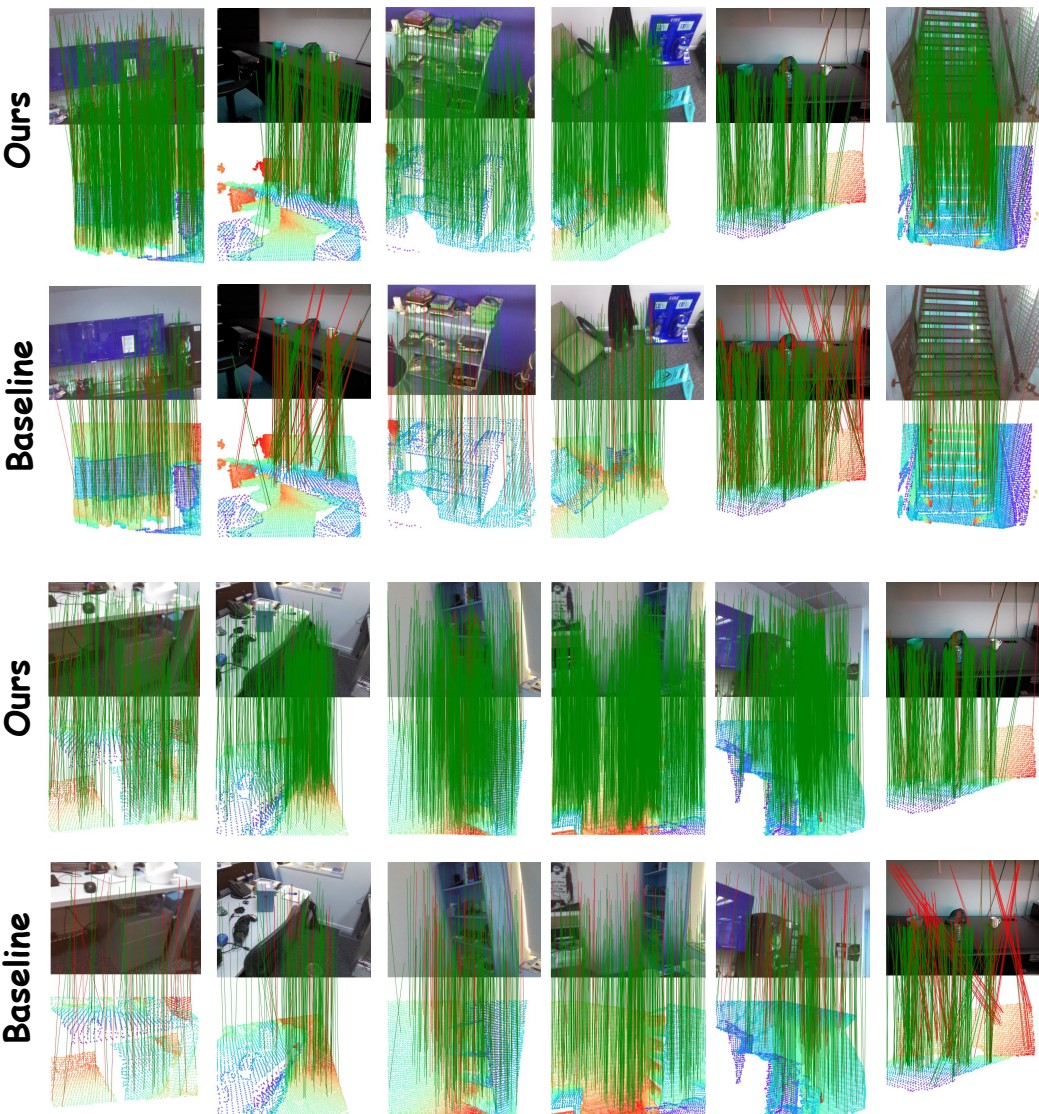

Figure 6: More visualization on Dataset.

## A.7 Addinoal Visualization

To further demonstrate the effectiveness of our approach, Figures 5 and 6 provide qualitative visualizations. Figure 5 shows projection-based visualizations, where point clouds are reprojected back onto the image plane using the estimated poses. Our approach consistently produces well-aligned projections with minimal misalignment across various indoor scenarios, including challenging cases with cluttered layouts, repetitive textures, and occlusions. These results further confirm that our method achieves more accurate and robust registration performance compared to the baseline.

In Figure 6, we compare our method with the baseline in terms of correspondence visualization. The green lines indicate correct matches and red lines represent incorrect ones. It can be observed that our method yields a much denser set of correct correspondences across different scenes, while the baseline often suffers from noticeable mismatches and large-scale deviations. This highlights the benefits of our overlap detection and domain alignment modules in improving correspondence reliability.

## A.8 Limitation and potential improvements

While our method significantly advances the state of image-to-point cloud registration, certain challenges, which have persisted in the field, remain. We have successfully addressed some of these issues, such as improving domain alignment and reducing mismatches, but there is still room for enhancement. In this section, we discuss these longstanding challenges and propose potential directions for future improvements that could further elevate the robustness and accuracy of our approach.

To gain a more comprehensive understanding of the limitations of our method and possible future improvement directions, we visualize several challenging cases. As shown in Figure 7, while our approach demonstrates clear improvements over previous baselines in these scenarios, some incorrect correspondences still persist. We observe that in regions with rapid depth changes, such as stairway gaps, ceiling vents, and hanger bases, the normals fail to provide reliable information and may even have adverse effects, leading to incorrect correspondences. We attempt to address this issue by proposing potential improvement directions. Since the ultimate goal of feature matching is to provide more reliable correspondences for the subsequent PnP-RANSAC algorithm, our efforts should focus on extracting correct correspondences.

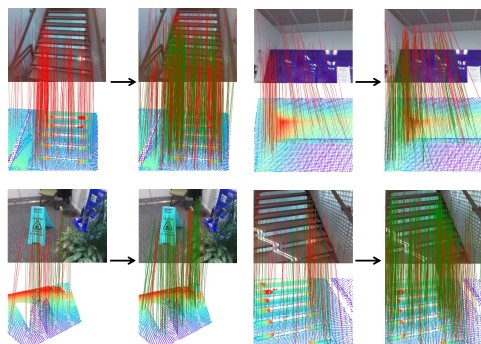

Figure 7: Limitation visualization on Dataset. The figure illustrates the comparison from the baseline to our method in challenging regions. The comparison from left to right represents the baseline and our method.

As a possible future direction, we plan to incorporate an uncertainty-aware local geometry modeling strategy. By explicitly modeling depth uncertainty, unreliable regions such as stair gaps or ceiling vents can be down-weighted, while local geometric fitting (e.g., plane or curvature regularization) can be used to refine normal estimation. This may alleviate the instability of normals in discontinuous regions and further reduce incorrect correspondences.

## A.9 Use of Large Models

In this work, large language models are employed solely for language polishing and improving the readability of the manuscript. They are not involved in problem formulation, algorithm design, model implementation, or experimental analysis. All technical contributions and experimental results are independently developed and verified by the authors.

