# OpenReview forum: "Learning Overlap Detection for Domain-Adaptive Image-to-Point Cloud Registration"
_ICLR.cc/2026/Conference — ICLR 2026 Conference Withdrawn Submission_

### Official Review · Reviewer_pubB · 2025-10-28

**Soundness:** 2
**Presentation:** 1
**Contribution:** 2
**Rating:** 2
**Confidence:** 4

**Summary:**

This paper addresses indoor image-to-point cloud registration by introducing two modules: (1) a Reinforcement Learning Overlap Detector (RLOD) that uses intrinsic geometric cues and adaptively identify overlapping regions, and (2) a Hierarchical Domain Adaptation Interaction (HDAI) module that aligns cross-modal features at mean and covariance levels. The method achieves state-of-the-art results on RGB-D Scenes v2 and 7-Scenes, outperforming prior methods.

**Strengths:**

The proposed method outperforms previous methods on two benchmarks by large margin

**Weaknesses:**

**1. Poor Presentation Quality**

The most critical issue with this paper is the poor quality of presentation, which significantly undermines its contributions.

a) Unclear Methodological Details

Critical implementation details are missing or inadequately explained:

- The paper fuses ResNet and DINOv2 features but provides neither motivation for this design choice nor technical details on how the fusion is performed. Such unsubstantiated methodological decisions weaken the paper's credibility.
- The Domain Adaptation Transformer (DAT) module lacks any technical specification. What are its inputs and outputs? How does it interact with the image feature pyramid as mentioned in lines 300-305? The paper provides no architectural details or implementation guidance.

b) Inconsistent Notations

The mathematical notation is inconsistent and poorly defined throughout:

- Equation 1: The variable $o$ (observation unit/patch pair) is used without clear definition of how patches are constructed from images and point clouds
- Equation 2: $f_x$  and $f_y$​ are undefined; it is unclear what x and y index
- Equations 1, 3, and 4: Cosine similarity is denoted inconsistently ($cos_f$) vs. $sim_{odm}$), while different distance metrics share the same notation ($sim_{odm}$ vs, $sim_{hist}$​), creating significant confusion
- Line 303: The image feature pyramid notation $f^2_{(i,n)}$​ is unexplained. What do the superscript 2, and subscripts i and n represent?

c) Writing Quality

The paper contains numerous grammatical errors and typos that detract from its professional quality:

- Line 213: "effectively reducing computational" lacks an object (should be "computational cost" or "computational complexity")
- Line 284: "reinferce" should be "reinforce"
- Section 4.4: Titled "Quantitative Analysis" but presents qualitative visualizations; should be "Qualitative Analysis"


d) Ambiguous Figures

Figure 2 (the main pipeline) contains unclear elements:

- What is the relationship between the "Domain Adapt Transformer" block on the left and the "DAT" module on the right? Are they the same component or different?
- What are the input features to the image feature pyramid, and where do they originate?
- What do the rocket symbols indicate?

**2. Unconvincing Motivation**

The introduction motivates the work by emphasizing the need for a "lightweight overlap detection module" to address computational complexity and efficiency concerns. However, the paper doesn't substantiates this claim. There is no discussion of computational efficiency in the main text. The appendix analysis (Tables 7-8) shows the proposed method actually consumes **more memory** (7942 MB vs. 6240 MB) and requires **longer computation time per iteration** (0.153s vs. 0.147s baseline) than the baseline. This contradicts the stated motivation and suggests that computational efficiency was not actually a primary design consideration.
The disconnect between the claimed motivation and actual results significantly undermines the paper's narrative coherence.

**3. Missing Experiments to Support Key Claims**

The paper makes several critical claims that lack experimental validation:

- Generalization capability: The authors claim their method "achieves excellent accuracy and strong generalization in cross-modal registration" (lines 104-106), yet only evaluate on two indoor datasets (RGB-D Scenes v2 and 7-Scenes), which share similar characteristics. No cross-dataset evaluation or testing on unseen environments is provided to substantiate the generalization claim.
- Attention drift mitigation: The paper asserts that the HDAI module "alleviates attention drift" and "stabilizes attention computation" (lines 107-109), but provides no ablation study or visualization demonstrating this effect. Without attention map visualizations or quantitative metrics showing reduced drift, this claim remains unsubstantiated.

These missing experiments leave key technical claims unverified and weaken confidence in the proposed contributions.

**Questions:**

1. The image encoder is trained to predict surface normals using predictions from Depth Anything v2 as pseudo ground-truth labels. Why train a separate predictor at all rather than directly using Depth Anything v2's predictions as input to the overlap detector?

---

### Official Review · Reviewer_Ksek · 2025-10-30

**Soundness:** 3
**Presentation:** 3
**Contribution:** 3
**Rating:** 4
**Confidence:** 5

**Summary:**

This paper performed a novel study on indoor image-to-point cloud registration by introducing a Reinforcement Learning Overlap Detector (RLOD) and a Hierarchical Domain Adaptation Interaction (HDAI) module, and demonstrated state-of-the-art performance on two standard benchmarks. This study contains some interesting findings and are valuable for the understanding of how to leverage intrinsic geometry and feature distribution alignment to address cross-modal discrepancies in registration tasks. However, lack of a thorough analysis on the computational overhead of the proposed RLOD module and some technical inconsistencies are the major flaws of the study.

**Strengths:**

1.This paper accurately identifies the registration differences between indoor and outdoor scenes. Proposing a lightweight RLOD module solves the core pain point of indoor scene overlap detection specialization.
2.The total loss function integrates matching loss, estimate normal loss, reinferce loss, and domain adaptation loss. The weight settings of each loss component have been experimentally verified, which can optimize the feature matching quality and cross modal alignment ability of the model.
3.Compared with mainstream methods on the three core indicators of IR, FMR, and RR, the proposed method shows significant performance advantages. Prove the domain alignment effect of HDAI module through t-SNE feature visualization, make the experimental conclusion is more convincing.

**Weaknesses:**

1.The Related Works section would benefit from including and discussing advancements in point cloud registration and related fields from the last two years (2024-2025), as the current references are predominantly from earlier periods, which does not fully reflect the most recent research context.
2.The visualization for the outdoor case (KITTI Odometry) in Fig. 1(a) appears to show a mismatch between the presented image and the point cloud, which undermines the credibility of the illustrative example.
3.A key inconsistency is the use of both regular L (e.g., in Eq. 13) and calligraphic \mathcal{L}(e.g., in Eq. 11, 12) to denote loss functions. The authors should adopt a single, consistent notation for all mathematical symbols.
4.The current numbering of equations in the manuscript is out of order, appearing as (1), (3), (4), (3), (5). This sequence is confusing and must be corrected to a logical and consecutive numbering scheme to ensure readability and proper citation.
5.The claim that the RLOD module is “lightweight” is not sufficiently supported. While Table 8 provides a high-level comparison, it does not isolate the overhead of RLOD. A granular analysis profiling the inference time and/or FLOPs of RLOD versus other components is necessary to validate this key claim.

**Questions:**

Please respond to the comments in Weaknesses. Please clarify ambiguities, supplement necessary evidence, and address existing limitations.

---

### Official Review · Reviewer_Douv · 2025-10-31

**Soundness:** 3
**Presentation:** 2
**Contribution:** 3
**Rating:** 6
**Confidence:** 4

**Summary:**

This paper presents a new framework for indoor image-to-point cloud registration by introducing two key modules: a Reinforcement Learning Overlap Detector (RLOD) for adaptive region selection and a Hierarchical Domain Adaptation Interaction (HDAI) module for feature distribution alignment. The method is evaluated on standard benchmarks (RGB-D Scenes v2 and 7-Scenes), demonstrating state-of-the-art performance across multiple metrics, with significant improvements in Registration Recall. The paper makes the following primary contributions:

It formulates the problem of overlap detection in indoor I2P registration as a reinforcement learning problem, leading to the RLOD module that adaptively selects candidate regions based on a fused geometric and appearance state vector.

It proposes the HDAI module, which integrates feature distribution alignment (mean and covariance) directly into the transformer's key space across multiple scales, effectively stabilizing cross-modal attention.

It demonstrates through extensive experiments that the combination of these modules achieves new state-of-the-art performance, with notable gains in challenging scenarios.

**Strengths:**

The application of RL to the specific problem of cross-modal overlap detection is creative. The state vector design, which combines intrinsic geometric invariants from normals with feature similarities, is a thoughtful and powerful representation for the policy network.

The HDAI module addresses a fundamental issue in cross-modal transformers—feature distribution mismatch. Aligning the mean and covariance of keys is a principled approach that directly mitigates "attention drift," and its hierarchical application across feature pyramid levels is a robust design.

The experimental section is a major strength. The authors go beyond standard metrics (IR, FMR, RR) to include relative pose errors (RRE, RTE) and comparisons with a wide range of methods, including zero-shot (FreeReg) and latest (Diff²I2P) approaches. The ablation studies clearly show the contribution of each component.

**Weaknesses:**

Clarity of Methodological Details: (1) The process by which candidate patch pairs are generated for RLOD is unclear. Is it an exhaustive comparison between all image and point cloud patches? If not, how is the candidate set pre-filtered? This crucial detail is missing. (2) The description of the "Hierarchical" nature of HDAI, while present, is scattered. A consolidated summary explicitly listing the different levels of hierarchy (statistical: mean/covariance; scale: image pyramid) would improve readability. (3) The circle loss (Eq. 12) is presented with complex, imported notation. A more intuitive explanation in the context of this paper's matching pipeline would be beneficial.

Limited Analysis of Generalization: The model demonstrates good within-domain generalization (unseen indoor scenes). However, the paper does not discuss or test its limits. How would the policy generalize to radically different indoor environments (e.g., industrial warehouses, churches) or outdoor scenes? An analysis or discussion of the learned policy's domain invariance would be valuable.

Reward Function Design: The reward function is a weighted sum of similarity metrics. While effective, the paper does not discuss potential alternatives or the sensitivity to the chosen weights (β_k). A brief discussion on the design choices and their robustness would strengthen the method.

**Questions:**

Could you please clarify the exact mechanism for generating the candidate patch pairs (x, y) that are evaluated by the RLOD policy network? Is it a dense matching between all patches, or is there a sampling strategy?

The reward function R is unsupervised. Did you experiment with incorporating any sparse, ground-truth supervised signals (e.g., from bilateral overlap) into the reward to further guide the policy, especially in the early stages of training?

The HDAI loss L_d is removed during inference. Did you observe any performance drop when doing so? Have you considered a test-time adaptation strategy to further boost performance on challenging test samples?

The learned RL policy is a single network for all scenes. Can you provide any analysis or visualization of what the policy has learned? For instance, does it prioritize different components of the state vector (e.g., geometry vs. appearance) in different types of scenes?

Typos and Minor Issues:
Page 3, Section 3: "prdict" should be "predict".

Page 6: Equation numbers are duplicated. Please re-number.

The references contain apparent duplicates (e.g., Huang et al. 2021a and 2021b are identical; Qin et al. 2022a, 2022b, 2022c are identical). Please check and correct.

---

### Official Review · Reviewer_szPA · 2025-11-01

**Soundness:** 2
**Presentation:** 2
**Contribution:** 2
**Rating:** 4
**Confidence:** 2

**Summary:**

This paper proposed a method for indoor image-to-point cloud registration. To minimizing interference from non-overlapping areas and reduce feature mismatches, they introduce reinforcement learning based detector and domain adaptation modules. The authors conduct experiments on RGB-D Scenes v2 and 7-Scenes datasets. Method are evaluated by Inlier Ratio, Feature Matching Recall and Registration Recall metrics. Compared to existing state-of-the-art approaches, the proposed method achieves state-of-the-art accuracy.

**Strengths:**

- state-of-the-art accuracy on indoor benchmarks.

**Weaknesses:**

- The proposed method is not clearly motivated — it is difficult to understand the author’s rationale for using reinforcement learning to train the detector.
- Moreover, the modality alignment is only addressed at the mean level, which makes it hard to demonstrate the method’s novelty. Since the author should be familiar with other alignment approaches, it would be better to include relevant works in the references.
- It is difficult to understand the author’s intent from some of the figures; for example, the rocket and leaf in Figure 2 are visually distracting.

**Questions:**

- What is the necessity of using reinforcement learning, and what advantages does it offer?

---

### Note · Authors · 2025-11-14

I have read and agree with the venue's withdrawal policy on behalf of myself and my co-authors.